# Optimization of monocrystalline silicon photovoltaic module assembly lines based on simulation model

Yuxiang Liu[1], Xinzhong Xia[2], Jingyang Zhang[1], Kun Wang[2], Bo Yu[3]*, Mengmeng Wu[2], Jinchao Shi[3], Chao Ma[2], Ying Liu[2], Boyang Hu[2], Xinying Wang[2], Bo Wang[1], Ruzhi Wang[1], Bing Wang[1]*

1 College of Materials Science and Engineering, Beijing University of Technology, Beijing, China, 2 Yingli Energy (China) Co., Ltd., Baoding, China, 3 National Key Laboratory of Photovoltaic Materials and Cells, Yingli Energy Development Co., Ltd., Baoding, China

☯ These authors contributed equally to this work.
* vincent.yu@yingli.com (BY); wangbing@bjut.edu.cn (BW)

## Abstract

This study presents a systematic approach to enhance the efficiency of monocrystalline silicon photovoltaic module assembly lines using advanced simulation modeling. The research focuses on developing a high-fidelity virtual model of the production line to replicate its physical layout, workflow sequences, and equipment interactions. Key assembly stages—including string welding, stacking, laminating, framing, and performance testing—are rigorously simulated to identify operational bottlenecks and inefficiencies. By analyzing workflow dynamics and resource utilization, targeted optimizations are proposed to streamline processes, reduce idle times, and improve throughput. Practical validation demonstrates that implementing these optimizations increases daily production output by over 6% and raises the production line balance rate by 5%, significantly lowering manufacturing costs while maintaining product quality. The methodology provides actionable insights for manufacturers to reconfigure production layouts, allocate resources effectively, and adapt to fluctuating market demands. This work bridges the gap between theoretical simulation and industrial implementation, offering a scalable framework for enhancing productivity, reducing waste, and advancing sustainable manufacturing practices in the photovoltaic sector. The findings highlight the critical role of simulation-driven strategies in addressing real-world engineering challenges and fostering cost-effective, high-efficiency production systems.

## Introduction

As a vital component of clean energy, the photovoltaic (PV) industry plays a crucial role in driving the transformation of the global energy structure and achieving the

**Data availability statement:** All relevant data are within the manuscript and its Supporting Information files.

**Funding:** The work was financially supported by the Baoding Science and Technology Plan Project (2394Z001). The funders had no role in study design, data collection and analysis, decision to publish, or preparation of the manuscript.

**Competing interests:** The authors have declared that no competing interests exist.

ambitious goal of carbon neutrality [1,2]. With the global commitment to reducing carbon emissions, the PV industry has emerged as a key area of development, attracting substantial investments and policy support. However, alongside these opportunities come significant challenges. The rapid expansion of the PV market, coupled with continuous technological advancements, has placed immense pressure on manufacturers to enhance production efficiency and control costs. Current production processes still exhibit inefficiencies in material consumption, energy utilization, and production yield, all of which directly impact the profitability and market competitiveness of PV enterprises. Addressing these inefficiencies is essential for maintaining the long-term sustainability and growth of the industry.

At the same time, the growing demand for diverse, customized products has driven the evolution of production methods. Small-batch, multi-species, and highly customized manufacturing are becoming mainstream, making product diversification, personalization, and customization increasingly prominent trends [3,4]. In response, the PV industry has progressively adopted advanced manufacturing paradigms that emphasize precision, intelligence, and sustainability. These paradigms are underpinned by the integration of virtual simulation technologies, which enable manufacturers to optimize production processes, enhance the intelligence of equipment, and reduce waste. Such approaches have become critical pathways for achieving high-efficiency and intelligent photovoltaic manufacturing in an increasingly competitive market environment.

Monocrystalline silicon photovoltaic modules represent a pivotal component in the solar PV manufacturing value chain. Their production process involves assembling monocrystalline silicon cell wafers into fully functional modules. As illustrated in Fig 1, the production line typically includes several sequential steps: string welding, stacking, laminating, framing, and inspection. The string welding process connects individual monocrystalline silicon cells into strings using soldering ribbons, ensuring electrical continuity. The stacking machine then layers these strings with encapsulation materials such as ethylene-vinyl acetate (EVA), backsheet, and tempered

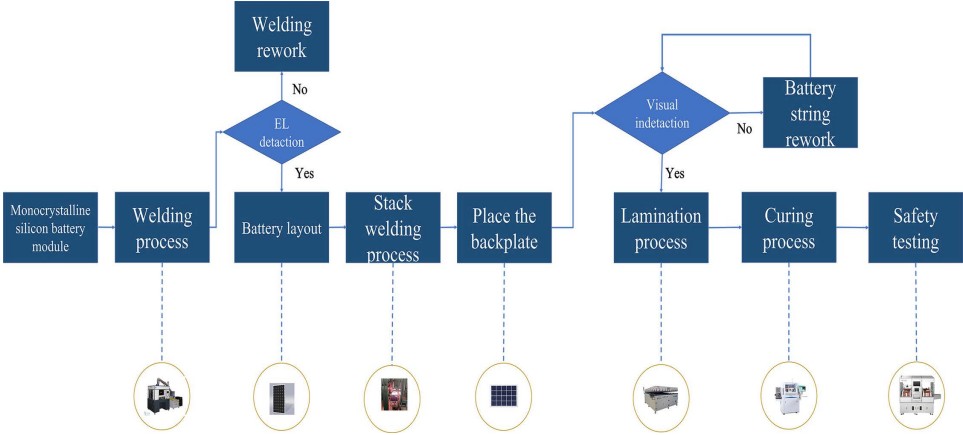

**Fig 1. Schematic diagram of monocrystalline silicon module assembly line.**

glass. The laminating step employs high temperatures to bond the encapsulation materials with the silicon cells, forming a robust, weather-resistant composite. Next, the framing machine assembles the laminated components into durable frames, enhancing mechanical strength and providing additional protection against environmental factors. Finally, the modules undergo rigorous electrical and optical performance tests, including power and electroluminescence (EL) testing, to verify their quality and compliance with industry standards.While conventional monocrystalline PV module assembly lines constitute mature manufacturing systems, their complex workflows and rigid production modes create significant bottlenecks in efficiency, cost-effectiveness, and operational flexibility. Current literature exhibits limited focus on PV module assembly line optimization, necessitating analytical emphasis on smart production systems employing full-cycle management through intelligent manufacturing approaches. Lean transformation and integrated optimization of production lines emerge as essential measures to address redundancy and inefficiency in smart manufacturing systems [5]. Although discrete optimization of production lines proves challenging for real-time implementation, digital twin (DT) technology offers novel solutions through comprehensive physical-digital mapping and multi-dimensional data integration for real-time simulation and optimization [6–10]. DT is a technology that enables comprehensive mapping and interaction with physical systems through digital models, combined with multidimensional information data to perform real-time simulation and optimization of systems [11–13]. DT provides modern solutions for optimizing production line systems and establishes theoretical models for handling complex process control in sustainable personalized manufacturing [14,15].

Recent years have witnessed significant advancements in digital simulation and optimization-driven production line design. Guo et al. employed (DT) technology to simulate balanced datasets for model training and transferred the trained models to physical production lines for fault diagnosis via transfer learning [16]. To evaluate system dynamic performance, Zhang et al. proposed a DT-based reconfiguration framework for semi-automated assembly lines, leveraging knowledge encapsulation techniques to facilitate virtual reconfiguration [17]. Dai et al. introduced a Smart Filter-assisted Domain Adversarial Neural Network (SFDANN) for fault diagnosis in noisy industrial environments [18]. Ghorvei et al. achieved unsupervised bearing fault diagnosis using a Deep Subdomain Adaptive Graph Convolutional Network (DSAGCN), integrating structured subdomain adaptation with domain adversarial learning [19]. Addressing label-free bearing fault diagnosis, Zhang et al. proposed a Collaborative Domain Adversarial Network (CDAN), enhancing diagnostic performance on unlabeled data through collaborative learning and adversarial mechanisms [20]. Li et al. established a DT-assisted framework for rolling bearing fault diagnosis under data imbalance conditions, combining synthetic data generated by DT with frequency-filtered subdomain adaptive networks [21]. Zhao et al. devised a wavelet-based DT-aided interpretable transfer learning framework, enabling intelligent fault diagnosis from simulated to real industrial domains by integrating DT with deep transfer learning techniques [22]. Wen et al. proposed a novel deep clustering network incorporating multi-representation autoencoders and adversarial learning, achieving large-scale cross-domain bearing fault diagnosis through enhanced clustering mechanisms [23]. However, previous studies predominantly focus on model-level optimization while neglecting explicit synchronization mechanisms between virtual and physical spaces, resulting in unreliable decision-making frameworks. To address dynamic disturbance impacts on production processes, this study proposes a coupled optimization approach considering system layout, resource scheduling, and process planning. We present a digital simulation-driven dynamic optimization methodology through physical-digital system verification, aiming to resolve multi-objective optimization challenges in monocrystalline PV module assembly lines. This approach seeks to enhance production throughput while improving economic efficiency, thereby contributing to sustainable advancement in PV manufacturing technologies.

This study proposes a DT-based simulation optimization method to enhance production efficiency and economic benefits in monocrystalline silicon photovoltaic module assembly lines. Addressing challenges in the photovoltaic industry such as efficiency bottlenecks, cost pressures, and diversified market demands during clean energy transition, the research focuses on dynamic production line optimization. A high-fidelity virtual simulation model is established to systematically resolve traditional assembly line issues including process redundancy, uneven resource allocation, and bottleneck process constraints.

Methodologically, the research initially constructs a digital model of a monocrystalline silicon module assembly line using Plant Simulation software, accurately replicating the physical workshop layout, equipment configuration, and process flow. Model validity is verified through real-world production data. Simulation analysis identifies the critical bottleneck process – the IV testing station in the detection area, which exhibits excessive workload (96.08%) and prolonged processing time (25 seconds), resulting in severe downstream congestion and a production line balance rate of merely 21.25%. To address this, an optimization strategy is implemented: reducing IV testing time to 20 seconds while applying ECRS principles (Eliminate, Combine, Rearrange, Simplify) for process improvement. Post-optimization simulation demonstrates a 15% workload reduction at IV testing stations, 6% daily output increase (from 6,637–7,038 units), and 5% improvement in line balance rate to 26.15%. Furthermore, comparative analysis of 10 scheduling rules (e.g., First-In-First-Out, Longest Processing Time) confirms that First-Come-First-Served rule maximizes total output, further validating method effectiveness.

The core contribution lies in integrating DT technology with dynamic optimization strategies, enabling real-time interaction and synchronous verification between virtual simulation and physical production. Through precise bottleneck identification, optimized resource scheduling, and balanced line loading, this approach significantly improves production efficiency while reducing manufacturing costs and enhancing adaptability to multi-batch customized production. Practical implementation achieves a 35,190-unit five-day output with notable economic benefits. This research provides a replicable technical framework for intelligent transformation in photovoltaic manufacturing and offers theoretical/practical references for similar discrete manufacturing scenarios.

## Digital modeling of production lines

### Purpose and process

The primary objective of production line modeling and simulation optimization is to analyze the performance of the production line, identify bottleneck stations that hinder its smooth operation, and enhance overall production efficiency. This is achieved by adjusting the processing times at bottleneck stations and optimizing associated work processes. The detailed workflow is illustrated in Fig 2.

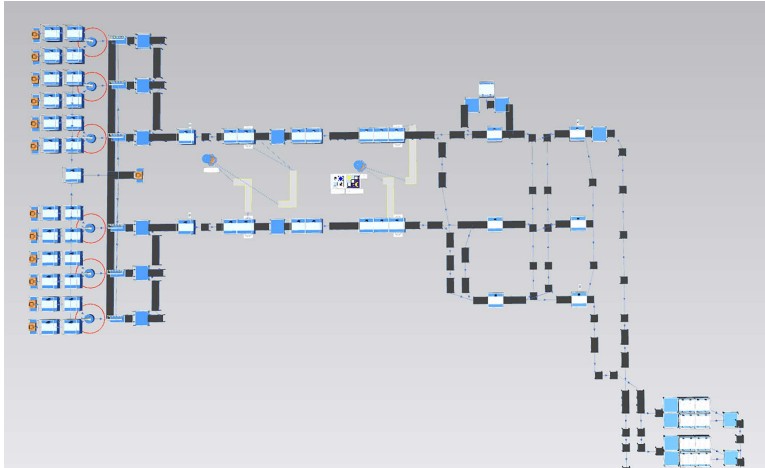

**Fig 2. Flow chart of modeling simulation.**

(1) System Analysis: Define the objectives of the simulation and gain a comprehensive understanding of the composition, structure, process flow, and equipment parameters of the production line. Collect real-time production data from equipment and workers to ensure accuracy and relevance.

(2) Model Construction: Develop a simulation model that replicates the physical layout of the factory, including workstations, equipment, warehouses, and transportation paths, using graphical tools. Integrate essential components into the model, such as production lines and transportation systems (e.g., conveyor belts, forklifts), and input the collected production data to configure resource parameters. Define the product flow path to establish a complete and functional model structure.

(3) Model Execution: Configure the start date and operational events for the simulation using the event manager. Execute and debug the model, collect simulation data, and prepare it for subsequent analysis.

(4) Correctness Verification: Process and analyze simulation data to validate the model's accuracy. Address inaccuracies through refinement or re-modeling. Utilize validated models and data to identify production line defects, pinpoint bottleneck stations, and provide reliable data for optimization.

(5) Optimization Analysis: Based on simulation results, propose an improvement plan following the ECRS principles (eliminate, combine, rearrange, simplify) to address identified inefficiencies.

(6) Evaluation of Optimization Effects: Compare pre- and post-optimization performance by assessing metrics such as output per unit time, enabling an evaluation of the effectiveness of the proposed optimization scheme.

## Data collection

This study takes the monocrystalline silicon module assembly production line of enterprise A as an example, in which the production line mainly includes a welding area, a stack welding area, a lamination curing area and a testing area, with a total of 12 welding machines, 2 stack welding machines, 8 laminating machines and two curing booths of large-scale equipment, and uses the stopwatch timing method to make 10 recordings in order to obtain the average processing time of the equipment of each work process. Considering that there are individual differences in the production time of workers, the normal distribution function is used for fitting and the outliers in the observed data are eliminated, and the average actual production time of each station of the production line is finally obtained as shown in Table 1.

## Software modeling

When using Plant Simulation software to model the assembly production line, the first step is to map the production elements to the software's entity types, using entities as the modeling basis to represent all production elements. Subsequently, process sequences are established according to the production timeline, and production elements are interconnected based on real-world operations. Finally, debugging and validation are performed to ensure the model's consistency with the actual production line. This modeling process enables simulation of the production line's operation and provides actionable optimization solutions. The mapping relationship between the production line's equipment entities and the modeled resource entities is shown in   Table 2.

## Tool selection and methodology

In this experiment, the Plant Simulation software was mainly chosen because of the following advantages:

• Industry-Specific Functionality: As a discrete-event simulation tool tailored for manufacturing systems, Plant Simulation is widely adopted in automotive and aerospace industries (cite relevant industry reports or literature). Its prebuilt libraries

for material flow logic, AGV routing, and resource optimization align directly with this study's objective of [insert specific objective, e.g., "identifying production line bottlenecks in high-mix manufacturing environments"].

- DT Compatibility: Unlike generic tools such as AnyLogic or Simio, Plant Simulation supports seamless integration with real-time IoT data streams (e.g., via OPC-UA protocols). This capability enables future scalability of the model into a DT framework, addressing a critical limitation of alternative platforms.

- Scenario Analysis Efficiency: The software's "Experiments" module facilitates automated multi-scenario testing with dynamic parameterization (e.g., batch size variability, shift schedule adjustments). This feature significantly reduces computational effort compared to manual scripting in Python-based simulators.

FlexSim and Arena were evaluated as alternatives. While FlexSim excels in 3D visualization, its limited support for complex routing logic rendered it unsuitable for modeling multi-stage rework loops. Similarly, Arena, though robust for

**Table 1. Actual production time at production line stations.**

| Work area | Station | Workstation operating time/s |
|---|---|---|
| Weld area | Welding machine | 10 |
| | String Check EL Detection | 1 |
| | Battery layout | 10 |
| Overlay zone | Stack welder | 17 |
| | Positioner | 9 |
| | Backsplash | 16 |
| Laminate curing zone | Laminating machine | 70 |
| | Placement of pads | 10 |
| | Backsplash | 16 |
| | Backplane | 15 |
| Inspection area | Upper and Lower Turning Station | 15 |
| | Appearance inspection station | 17 |
| | IV Testing | 25 |
| | EL Detection | 17 |
| | Labeling station | 13 |
| | Corner wrapping station | 15 |
| | Safety Testing station | 17 |
| | Adapter removal station | 15 |

**Table 2. Mapping relationship between production line entities and model entities.**

| Production line entities | Simulation Entity Library Objects | Quantity |
|---|---|---|
| Welding Machines | Processor | 12 |
| Nesting Machine | Processor | 6 |
| Stacking machine | Processor | 2 |
| Cutting Machine | Processor | 10 |
| EL Tester | Processor | 15 |
| Laminating Machine | Processor | 8 |
| IV Tester | Processor | 2 |
| Staging Area | Memory | 25 |
| Conveyor Belt | Transporter | 100 |

service systems, lacks dedicated modules for plant layout optimization. By contrast, Plant Simulation's Flow Control Language provided granular control over priority-based buffer allocation, which was essential for validating our hypothesis.

Methodology Justification: Initial time studies and process mapping were necessary because factory lacked digitized MES records. We followed ISO 22400 standards for measurement consistency, with inter-rater reliability tests (Cohen's $\kappa = 0.82$).

## Results and discussion

### Simulation modeling

One of the key components of production line simulation is the creation of an accurate and representative model. In this study, the simulation model was developed based on the actual production line layout within the manufacturing workshop of Company A. Using Plant Simulation software, a detailed functional model of the production line was constructed. The two-dimensional representation of the production line is depicted in Fig 3, illustrating its primary processing areas. These areas include:

- A welding section comprising 12 welding machines;

- A stack welding section equipped with 2 stack welding machines;

- A film processing section consisting of 2 positional glue machines, 2 manually operated pad placement stations, and 2 backside film machines.

To further enhance the accuracy and realism of the simulation, a three-dimensional physical model of the production line was also developed using Plant Simulation software, as shown in Fig 3. This 3D model replicates the factory's production line at a 1:1 scale, ensuring that all units, components, and spatial arrangements are faithfully reproduced. The realistic visualization offered by the 3D model facilitates a comprehensive understanding of the production workflow, enabling detailed analysis and optimization of each production stage. By accurately representing the physical workshop environment, the model serves as a powerful tool for identifying inefficiencies, testing potential improvements, and supporting decision-making processes in production line management.

### Identification and optimization of bottleneck processes in production lines

The bottleneck process is the process that has the longest operating time in the production system, and its long elapsed time seriously affects the production efficiency of the enterprise. In order to improve efficiency, bottleneck processes must

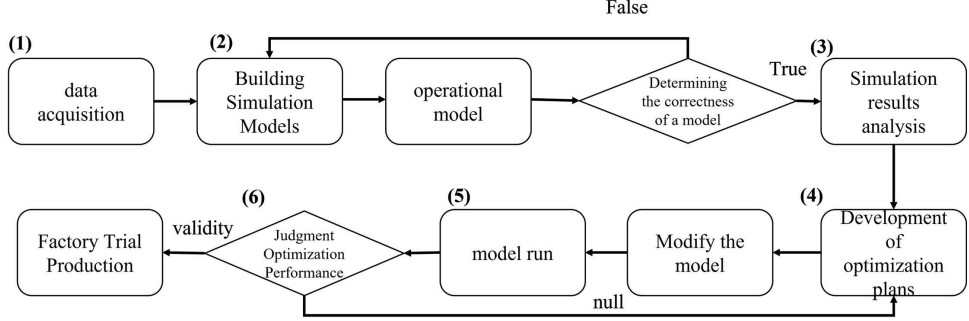

**Fig 3. Production line simulation model.**

---

be optimized. In the simulation analysis, the main focus is on the simulation data to identify and suggest optimization of the bottleneck process. The bottleneck process is identified in the following way:

$$A = \max\left(\frac{1}{m}\sum_{1}^{m}Q_{1j} + T_{1j}, \frac{1}{n}\sum_{1}^{n}Q_{2j} + T_{2j}, \ldots, \frac{1}{q}\sum_{1}^{q}Q_{ij} + T_{ij}\right)$$

(1)

[i] - Process sequences
[j] - Equipment sequences
[m.n…q] - Number of equipment for the process
[$Q_{ij}$] - The processing time of process i on equipment j
[$T_{ij}$] - The residence time in the buffer zone before and after the process

Using Plant Simulation software, the simulation model was established and executed under the configured conditions. Through analysis, the bottleneck station was identified in the inspection area, with a total output of 33,185 units over 5 days (average daily output: 6,637 units). Bottleneck analysis conducted at the simulation conclusion revealed key metrics from the original inspection area results, as shown in Table 3.

As detailed in Table 3, the IV test station exhibited the highest utilization rate (92.4%) among all stations, operating under critical workload conditions and emerging as the primary production line bottleneck. In contrast, other stations demonstrated substantially lower utilization rates, with notable congestion at the turnover stations (19.3% blockage rate) and the appearance inspection station (30.1% blockage rate). This resource allocation imbalance resulted in suboptimal production line efficiency, yielding a production line balance rate of only 21.25%.

To address this constraint, an experimental design was implemented by incrementally adjusting the IV test station's processing speed. Simulation tests evaluated cycle times ranging from 15 to 25 seconds, monitoring both total yield and downstream station performance metrics. The results in Table 4 demonstrate that reducing the IV test station's processing

**Table 3. Analysis of bottleneck data before optimization.**

| Station | Working | Waiting | Blocking | Sorting |
|---|---|---|---|---|
| (A1)IV Testing12 | 96.08 | 3.92 | 0 | 96.08 |
| (A2)IV Testing13 | 96.02 | 3.98 | 0 | 96.02 |
| (B1)Upper and Lower Turning Station | 57.66 | 4.04 | 38.30 | 57.66 |
| (B2)Upper and Lower Turning Station2 | 57.65 | 42.35 | 0 | 57.65 |
| (B3)Upper and Lower Turning Station 1 | 57.63 | 4.05 | 38.33 | 57.63 |
| (B4)Upper and Lower Turning Station 4 | 57.61 | 42.39 | 0 | 57.61 |
| (C1)Appearance inspection station | 65.34 | 4.00 | 30.66 | 65.34 |
| (C2)Appearance inspection station 11 | 65.30 | 4.03 | 30.67 | 65.30 |
| (D1)Safety Testing station | 65.33 | 34.67 | 0 | 65.33 |
| (D2)Safety Testing station1 | 65.29 | 34.71 | 0 | 65.29 |
| (E1)EL Detection 4 | 65.29 | 0 | 34.71 | 65.29 |
| (E2)EL Detection 3 | 65.33 | 0 | 34.67 | 65.33 |
| (F1)Adapter removal station | 57.64 | 42.36 | 0 | 57.64 |
| (F2)Adapter removal station 1 | 57.60 | 42.40 | 0 | 57.60 |
| (G1)Corner wrapping station | 57.63 | 42.37 | 0 | 57.63 |
| (G2)Corner wrapping station 1 | 57.59 | 42.41 | 0 | 57.59 |
| (H1)Labeling station | 49.95 | 50.05 | 0 | 49.95 |
| (H2)Labeling station 2 | 49.92 | 50.08 | 0 | 49.92 |

**Table 4. Optimization of the experimental procedure.**

| IV test processing time | IV Testing12 | | IV Testing13 | | Yield |
|---|---|---|---|---|---|
| | Working | Waiting | Working | Waiting | |
| 15 | 61.12 | 38.88 | 61.10 | 38.90 | 35186 |
| 16 | 65.19 | 34.81 | 65.17 | 34.83 | 35185 |
| 17 | 69.27 | 30.73 | 69.25 | 30.75 | 35185 |
| 18 | 73.34 | 26.66 | 73.32 | 26.68 | 35185 |
| 19 | 77.42 | 22.58 | 77.39 | 22.61 | 35185 |
| 20 | 81.49 | 18.51 | 81.47 | 18.53 | 35185 |
| 21 | 85.57 | 14.43 | 85.51 | 14.49 | 35178 |
| 22 | 89.94 | 10.36 | 89.49 | 10.51 | 35162 |
| 23 | 93.71 | 6.29 | 93.48 | 6.52 | 35146 |
| 24 | 96.07 | 3.93 | 95.96 | 4.04 | 34553 |
| 25 | 96.08 | 3.92 | 96.02 | 3.98 | 33182 |

time correlated with increased production line throughput. When processing time reached 20 seconds, the total yield plateaued at 35,190 units over 5 days, indicating bottleneck elimination at this station.

Optimizing the IV test station to a 20-second cycle time delivered significant improvements:

• Total output increased by 6.05% (7,038 units/day vs. original 6,637 units/day)

• Production line balance rate rose to 26.15% (4.9 percentage point improvement)

Post-optimization bottleneck data in Table 5 shows a 15.2% reduction in the IV test station's utilization rate, along with improved workload distribution across inspection area stations. Fig 4 visually contrasts resource utilization patterns before and after optimization, highlighting enhanced workload equilibrium.

**Table 5. Analysis of bottleneck data after optimization.**

| Station | Working | Waiting | Blocking | Sorting |
|---|---|---|---|---|
| (A1)IV Testing 12 | 81.49 | 18.51 | 0 | 81.49 |
| (A2)IV Testing 13 | 81.47 | 18.53 | 0 | 81.47 |
| (B1)Upper and Lower Turning Station | 61.13 | 38.87 | 0 | 61.13 |
| (B2)Upper and Lower Turning Station2 | 61.12 | 38.88 | 0 | 61.12 |
| (B3)Upper and Lower Turning Station1 | 61.11 | 38.89 | 0 | 61.11 |
| (B4)Upper and Lower Turning Station4 | 61.10 | 38.90 | 0 | 61.10 |
| (C1)Appearance inspection station | 69.27 | 30.73 | 0 | 69.27 |
| (C2)Appearance inspection station11 | 69.25 | 30.75 | 0 | 69.25 |
| (D1)Safety Testing station | 69.26 | 30.74 | 0 | 69.26 |
| (D2)Safety Testing station 1 | 69.24 | 30.76 | 0 | 69.24 |
| (E1)EL Detection 4 | 69.24 | 30.76 | 0 | 69.24 |
| (E2)EL Detection 3 | 69.26 | 30.74 | 0 | 69.26 |
| (F1)Adapter removal station | 61.11 | 38.89 | 0 | 61.11 |
| (F2)Adapter removal station 1 | 61.09 | 38.91 | 0 | 61.09 |
| (G1)Corner wrapping station | 61.10 | 38.90 | 0 | 61.10 |
| (G2)Corner wrapping station 1 | 61.08 | 38.92 | 0 | 61.08 |
| (H1)Labeling station | 52.95 | 47.05 | 0 | 52.95 |
| (H2)Labeling station 2 | 52.94 | 47.06 | 0 | 52.94 |

This intervention demonstrates that targeted cycle time reduction effectively mitigates bottleneck constraints, enhancing both production efficiency (throughput +6%) and system balance (balance rate +23%). The resulting workflow optimization contributes to sustainable operational gains and increased enterprise profitability.

**Validation status**

While the simulation results demonstrate significant improvements in production efficiency (6% output increase and 5% balance rate enhancement), it should be noted that:

- Current Stage: The optimization outcomes are based on digital simulation validated against historical production data (2023 average: 6512 units/day vs simulated 6637 units/day, deviation: 1.9%).

- Physical Implementation:

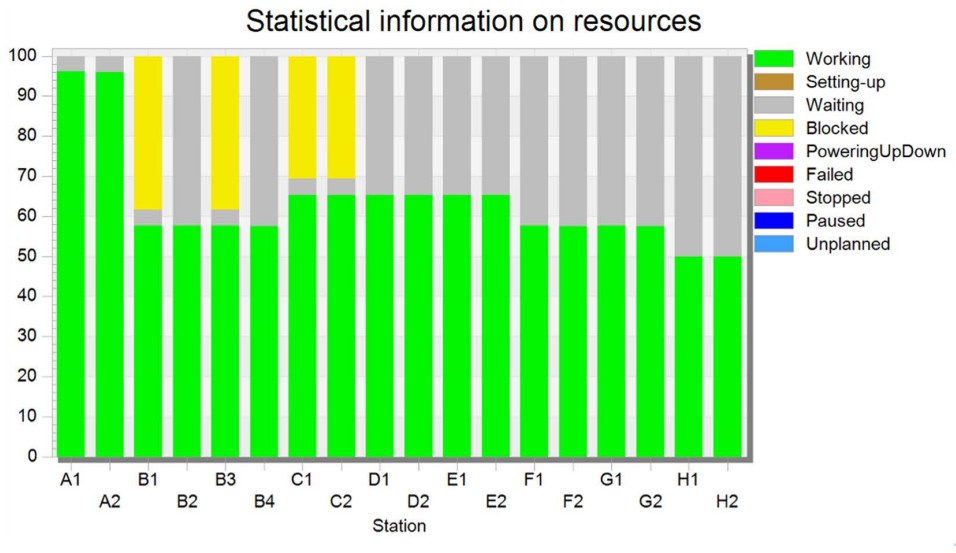

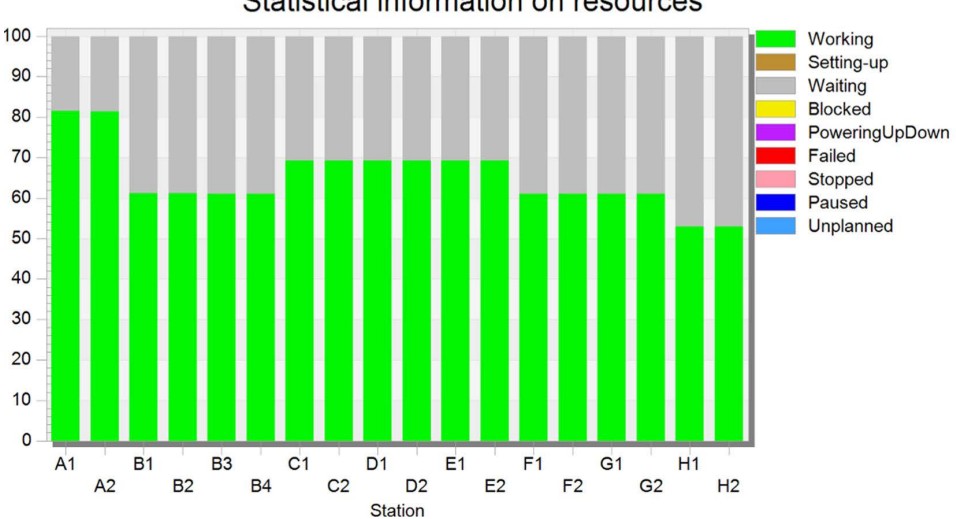

**Fig 4. Resource information graph before and after optimization.**

(1) The proposed 20s cycle time at IV test station requires hardware upgrades (firmware update + parallel processing module).

(2) Factory acceptance tests are scheduled for Q3 2025 due to necessary production line shutdowns.

• Next Steps:

(1) Full-scale production validation will be conducted post-hardware modification.

(2) Results will be reported in a subsequent industrial case study.

## Conclusion

With the continuous growth of market demand in the photovoltaic industry, establishing a high productivity and low cost production assembly line has become essential to cope with industrial challenges. This paper takes the assembly process of monocrystalline silicon cell module of Company A as the research object and introduces its main assembly process. Based on the DT technology, a workshop optimization model in line with the logic of production line construction and production capacity per unit area is established, and the proposed scheme is simulated and verified using DT semi-physical simulation technology. The final production line balance rate is increased by more than 5%, and the production capacity is increased by more than 6% in the same time, which saves a lot of production cost for the enterprise and improves the economic efficiency of the enterprise. The above research provides solutions and technical solutions for the design and optimization of monocrystalline silicon cell module assembly plant with multiple batches and high-frequency production variations. It is also useful for the design of similar products.

## Supporting information

**S1 File. Processing time of bottleneck equipment and the working data of bottleneck equipment before and after optimization.**
(ZIP)

## Author contributions

**Data curation:** Mengmeng Wu, Jinchao Shi, Chao Ma, Xinying Wang.

**Formal analysis:** Jingyang Zhang.

**Funding acquisition:** Xinzhong Xia, Bing Wang.

**Investigation:** Jingyang Zhang, Kun Wang.

**Methodology:** Yuxiang Liu.

**Project administration:** Xinzhong Xia.

**Supervision:** Bo Yu, Bo Wang.

**Validation:** Yuxiang Liu, Ying Liu.

**Visualization:** Boyang Hu.

**Writing – original draft:** Yuxiang Liu.

**Writing – review & editing:** Ruzhi Wang.

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
