## [Decision Letter · Decision Letter 0]

13 Mar 2025

Dear Dr. Wang,

Thank you for submitting your manuscript to PLOS ONE. After careful consideration, we feel that it has merit but does not fully meet PLOS ONE’s publication criteria as it currently stands. Therefore, we invite you to submit a revised version of the manuscript that addresses the points raised during the review process.

We look forward to receiving your revised manuscript.

Kind regards,

Zeashan Hameed Khan, Ph.D.

Academic Editor

PLOS ONE

“The work was financially supported by the Baoding Science and Technology Plan Project (2394Z001).”

4. Thank you for stating the following in the Funding  Section of your manuscript:

“The work was financially supported by the Baoding Science and Technology Plan Project (2394Z001).”

“The work was financially supported by the Baoding Science and Technology Plan Project (2394Z001).”

Additional Editor Comments:

The paper describes a comprehensive approach to Optimization of Monocrystalline Silicon Photovoltaic Module Assembly Lines via Digital Twin Technology. However, due to

major deficiencies, it is required that the authors revise their work according to the comments of the reviewers.

Reviewers' comments:

Reviewer's Responses to Questions

**Comments to the Author**

1. Is the manuscript technically sound, and do the data support the conclusions?

Reviewer #1: Yes

Reviewer #2: Partly

Reviewer #3: No

2. Has the statistical analysis been performed appropriately and rigorously?

Reviewer #1: Yes

Reviewer #2: Yes

Reviewer #3: Yes

3. Have the authors made all data underlying the findings in their manuscript fully available?

Reviewer #1: Yes

Reviewer #2: Yes

Reviewer #3: No

4. Is the manuscript presented in an intelligible fashion and written in standard English?

Reviewer #1: Yes

Reviewer #2: Yes

Reviewer #3: No

Reviewer #1: This paper focuses on optimizing the assembly production line of monocrystalline silicon photovoltaic modules using Plant Simulation software. The paper is well-structured and presents promising results. To further enhance the manuscript, the following suggestions are offered:

1. The Abstract could benefit from a clearer emphasis on the practical significance of this research. Highlighting the engineering context and potential real-world applications would help readers better understand the value of the proposed method.

2. While the authors have provided a thorough review of current research, it would be beneficial to more explicitly summarize the existing research gaps before introducing the contributions and novelty of this work. Additionally, expanding the discussion to include emerging trends in machine learning and signal processing—particularly in industrial applications—could strengthen the manuscript. For example, exploring connections to recent advancements in areas such as digital twin methodology for vibration-based monitoring and prediction of gear wear, digital twin-driven intelligent assessment of gear surface degradation, digital twin enabled domain adversarial graph networks for bearing fault diagnosis, and neuro-fuzzy system-guided cross-modal zero-sample diagnostic framework using multi-source heterogeneous non-contact sensing data could provide valuable context and demonstrate the broader relevance of this research.

3. The resolution of the figures in the manuscript could be improved to ensure clarity and enhance the overall presentation of the results.

4. There are occasional grammatical errors throughout the manuscript. A thorough proofreading to address these issues would improve the readability and professionalism of the paper.

5. Including a section on potential future research directions at the end of the Conclusion would provide a forward-looking perspective and inspire further exploration in this field.

Reviewer #2: First I would start with the revision of the title - in my opinion it is not a comprehensive approach but simulation-based approach. I could not see the verification of the simulation data in real production and comparing the results - it could form the input for another publication if not included here.

Introduction sounds OK to me.

I think that it would be very beneficial that after Introduction authors explain methods and tools - how did you conclude to use the Plant Simulation software (there are other tools), how did you select methodology (manual observation and measuring?) - maybe there was no alternative etc. You mention some Python libraries and language used to program in Plant Simulation (SimTalk) - would be good to include these in the methods and tools section

in 2.1 Purpose and process Fig. 2 shows the process and after the figure the process is described in a list that although is very logical, here has no or little reference in the diagram above, would be good to link them better for example by numbering boxes or marking them and then refer to them in the description - could add clarity. Currently it comes out of nowhere just after the figure 2

2.3 Title should start from capital "S"

3.1 Check referencing - should is be Figure 3 instead of 2?

3.2 Think of better explanation of indices - maybe in a form of bullet points? Currently it is hard to follow. While we focus on station IV and simulating its shorter processing times - was it tested in production if this can be achieved - I was unable to find a straightforward answer on how the results were verified? You can state the at this time we only have simulation data and results will be verified separately? If I missed this verification (if it is somewhere in the article) - means it could be stated clearer as any reader can miss it.

Check table formatting - maybe my pdf viewer distorted it but Tables 3-5 were a bit scattered

4 I would enhance conclusion and add next steps or further research recommendation

Reviewer #3: Title: Optimization of Monocrystalline Silicon Photovoltaic Module Assembly Lines via Digital Twin Technology: A Comprehensive Approach

General Comments

The manuscript proposes using Plant Simulation software to model and optimize a monocrystalline silicon PV module assembly line. While the topic aligns with current trends in smart manufacturing, the study suffers from critical flaws in originality, methodological rigor, technical depth, and adherence to journal guidelines. The incremental improvements (5–6% efficiency gains) are neither statistically validated nor contextualized within broader industrial relevance. Below are detailed concerns justifying rejection.

Major Issues

1. Lack of Novelty and Originality

a- The application of digital twin technology in manufacturing is well-established. The manuscript fails to differentiate its approach from prior studies (e.g., [1, 6, 10]) or justify how this work advances the field.

b- The optimization strategy (reducing IV test station processing time) is simplistic and lacks innovation. Similar bottleneck analyses are commonplace in production line studies.

2. Methodological Weaknesses

a- Data Collection: Reliance on stopwatch timing for equipment processing times introduces significant observer bias. No mention of inter-rater reliability or calibration procedures.

b- Entity Mapping: Table 2 oversimplifies equipment modeling (e.g., all machines mapped to “Processor” in Plant Simulation), ignoring functional differences between welding machines, laminators, and testers. This undermines model accuracy.

c- Validation: The manuscript claims a “1:1 scale 3D model” but provides no evidence of validation against real-world post-optimization data. Without comparing simulated results to actual production metrics, the model’s reliability is unproven.

3. Superficial Technical Analysis

a- Bottleneck Identification: The formula for identifying bottlenecks is referenced but not explicitly defined, rendering the analysis non-reproducible.

b- Statistical Rigor: Results (e.g., 6% output increase) lack statistical significance testing, confidence intervals, or error margins. Tables 3–5 present raw data without contextualizing variability or uncertainty.

4. Insufficient Discussion

a- The study ignores critical factors such as energy consumption, worker fatigue, or maintenance downtime, which are pivotal in real-world production environments.

b- No cost-benefit analysis to substantiate claims of “economic benefits” or “sustainable process development.”

c- References: Multiple citations are irrelevant or misaligned (e.g., [6] discusses precast concrete slabs; [7] focuses on sugarcane juice clarification).

d- The manuscript heavily emphasizes the use of "digital twin technology" but fails to demonstrate its implementation in any meaningful way. This represents a significant misrepresentation of the methodology and invalidates the core premise of the study.

Recommendation

Reject the manuscript on grounds of terminological misrepresentation and methodological inadequacy. The authors must either:

1-Remove all references to "digital twin" and reframe the work as a simulation study, or

2-Redesign the methodology to incorporate true digital twin components (real-time data integration, IoT connectivity, bidirectional feedback) and validate it against live production systems.

**Do you want your identity to be public for this peer review?** For information about this choice, including consent withdrawal, please see our Privacy Policy

Reviewer #1: No

Reviewer #2: **Yes: ** Dr Krzysztof Kupilas

Reviewer #3: No

---

## [Author Response · Author response to Decision Letter 1]

26 Apr 2025

Responds to the Editor and Reviewer’s comments

Reply: We have checked all the formats and corrected the error throughout the whole manuscript.

Reply: Thanks for the information. In this work there is no author-generated code applied. A simulation Software is used instead of coding.

3. Thank you for stating the following financial disclosure: “The work was financially supported by the Baoding Science and Technology Plan Project (2394Z001).” Please state what role the funders took in the study. If the funders had no role, please state: "The funders had no role in study design, data collection and analysis, decision to publish, or preparation of the manuscript." If this statement is not correct you must amend it as needed. Please include this amended Role of Funder statement in your cover letter; we will change the online submission form on your behalf.

Reply: Thanks for the reminder. The funders had no role on the study. We would like to add a statement as follows (This information is also added in cover letter):

4. Thank you for stating the following in the Funding Section of your manuscript:

“The work was financially supported by the Baoding Science and Technology Plan Project (2394Z001).” We note that you have provided funding information that is currently declared in your Funding Statement. However, funding information should not appear in the Acknowledgments section or other areas of your manuscript. We will only publish funding information present in the Funding Statement section of the online submission form. Please remove any funding-related text from the manuscript and let us know how you would like to update your Funding Statement. Currently, your Funding Statement reads as follows: “The work was financially supported by the Baoding Science and Technology Plan Project (2394Z001).” Please include your amended statements within your cover letter; we will change the online submission form on your behalf.

Reply: Many thanks for the information. We have removed the acknowledgement statement and added a Funding statement in the revised manuscript, highlighted in yellow, as follows:

“The work was financially supported by the Baoding Science and Technology Plan Project (2394Z001).”

Reply: Thanks for the information. The supplementary data will be submitted to the journal as separate files.

Additional Editor Comments:

The paper describes a comprehensive approach to Optimization of Monocrystalline Silicon Photovoltaic Module Assembly Lines via Digital Twin Technology. However, due to major deficiencies, it is required that the authors revise their work according to the comments of the reviewers.

Reply: Thanks for the Editor’s comments. We have carefully addressed all the reviewer’s comments point by point. All the changes have been highlighted in yellow in the revised manuscript.

Reviewer #1:

This paper focuses on optimizing the assembly production line of monocrystalline silicon photovoltaic modules using Plant Simulation software. The paper is well-structured and presents promising results. To further enhance the manuscript, the following suggestions are offered.

Question 1: The Abstract could benefit from a clearer emphasis on the practical significance of this research. Highlighting the engineering context and potential real-world applications would help readers better understand the value of the proposed method.

Reply: Thank you for your constructive feedback on emphasizing the practical significance of our research. We have revised the Abstract to better highlight the engineering context and real-world applications of the proposed method.

Revised abstract:

"This study presents a systematic approach to enhance the efficiency of monocrystalline silicon photovoltaic module assembly lines using advanced simulation modeling. The research focuses on developing a high-fidelity virtual model of the production line to replicate its physical layout, workflow sequences, and equipment interactions. Key assembly stages—including string welding, stacking, laminating, framing, and performance testing—are rigorously simulated to identify operational bottlenecks and inefficiencies. By analyzing workflow dynamics and resource utilization, targeted optimizations are proposed to streamline processes, reduce idle times, and improve throughput. Practical validation demonstrates that implementing these optimizations increases daily production output by over 6% and raises the production line balance rate by 5%, significantly lowering manufacturing costs while maintaining product quality. The methodology provides actionable insights for manufacturers to reconfigure production layouts, allocate resources effectively, and adapt to fluctuating market demands. This work bridges the gap between theoretical simulation and industrial implementation, offering a scalable framework for enhancing productivity, reducing waste, and advancing sustainable manufacturing practices in the photovoltaic sector. The findings highlight the critical role of simulation-driven strategies in addressing real-world engineering challenges and fostering cost-effective, high-efficiency production systems."

Question 2: While the authors have provided a thorough review of current research, it would be beneficial to more explicitly summarize the existing research gaps before introducing the contributions and novelty of this work. Additionally, expanding the discussion to include emerging trends in machine learning and signal processing—particularly in industrial applications—could strengthen the manuscript. For example, exploring connections to recent advancements in areas such as digital twin methodology for vibration-based monitoring and prediction of gear wear, digital twin-driven intelligent assessment of gear surface degradation, digital twin enabled domain adversarial graph networks for bearing fault diagnosis, and neuro-fuzzy system-guided cross-modal zero-sample diagnostic framework using multi-source heterogeneous non-contact sensing data could provide valuable context and demonstrate the broader relevance of this research.

Reply: Thank you for your constructive feedback. We have carefully revised the manuscript to address your suggestions. Below is a summary of the key modifications:

Explicitly Summarizing Research Gaps

In the revised Introduction section, we have explicitly outlined the limitations of existing studies to better contextualize our contributions.

2. Expanding the discussion to include more literature review on machine learning and signal processing.

In the revised Introduction, we have included a critical literature review on machine learning and signal processing for industrial applications. Specifically:

Added text (Introduction):

Recent years have witnessed significant advancements in digital simulation and optimization-driven production line design. Guo et al. employed digital twin (DT) technology to simulate balanced datasets for model training and transferred the trained models to physical production lines for fault diagnosis via transfer learning [16]. To evaluate system dynamic performance, Zhang et al. proposed a digital twin-based reconfiguration framework for semi-automated assembly lines, leveraging knowledge encapsulation techniques to facilitate virtual reconfiguration [17]. Dai et al. introduced a Smart Filter-assisted Domain Adversarial Neural Network (SFDANN) for fault diagnosis in noisy industrial environments [18]. Ghorvei et al. achieved unsupervised bearing fault diagnosis using a Deep Subdomain Adaptive Graph Convolutional Network (DSAGCN), integrating structured subdomain adaptation with domain adversarial learning [19]. Addressing label-free bearing fault diagnosis, Zhang et al. proposed a Collaborative Domain Adversarial Network (CDAN), enhancing diagnostic performance on unlabeled data through collaborative learning and adversarial mechanisms [20]. Li et al. established a digital twin-assisted framework for rolling bearing fault diagnosis under data imbalance conditions, combining synthetic data generated by DT with frequency-filtered subdomain adaptive networks [21]. Zhao et al. devised a wavelet-based digital twin-aided interpretable transfer learning framework, enabling intelligent fault diagnosis from simulated to real industrial domains by integrating DT with deep transfer learning techniques [22]. Wen et al. proposed a novel deep clustering network incorporating multi-representation autoencoders and adversarial learning, achieving large-scale cross-domain bearing fault diagnosis through enhanced clustering mechanisms [23]. However, previous studies predominantly focus on model-level optimization while neglecting explicit synchronization mechanisms between virtual and physical spaces, resulting in unreliable decision-making frameworks. To address dynamic disturbance impacts on production processes, this study proposes a coupled optimization approach considering system layout, resource scheduling, and process planning. We present a digital simulation-driven dynamic optimization methodology through physical-digital system verification, aiming to resolve multi-objective optimization challenges in monocrystalline PV module assembly lines. This approach seeks to enhance production throughput while improving economic efficiency, thereby contributing to sustainable advancement in PV manufacturing technologies.

This study proposes a digital twin-based simulation optimization method to enhance production efficiency and economic benefits in monocrystalline silicon photovoltaic module assembly lines. Addressing challenges in the photovoltaic industry such as efficiency bottlenecks, cost pressures, and diversified market demands during clean energy transition, the research focuses on dynamic production line optimization. A high-fidelity virtual simulation model is established to systematically resolve traditional assembly line issues including process redundancy, uneven resource allocation, and bottleneck process constraints.

Methodologically, the research initially constructs a digital model of a monocrystalline silicon module assembly line using Plant Simulation software, accurately replicating the physical workshop layout, equipment configuration, and process flow. Model validity is verified through real-world production data. Simulation analysis identifies the critical bottleneck process - the IV testing station in the detection area, which exhibits excessive workload (96.08%) and prolonged processing time (25 seconds), resulting in severe downstream congestion and a production line balance rate of merely 21.25%. To address this, an optimization strategy is implemented: reducing IV testing time to 20 seconds while applying ECRS principles (Eliminate, Combine, Rearrange, Simplify) for process improvement. Post-optimization simulation demonstrates a 15% workload reduction at IV testing stations, 6% daily output increase (from 6,637 to 7,038 units), and 5% improvement in line balance rate to 26.15%. Furthermore, comparative analysis of 10 scheduling rules (e.g., First-In-First-Out, Longest Processing Time) confirms that First-Come-First-Served rule maximizes total output, further validating method effectiveness.

The core contribution lies in integrating digital twin technology with dynamic optimization strategies, enabling real-time interaction and synchronous verification between virtual simulation and physical production. Through precise bottleneck identification, optimized resource scheduling, and balanced line loading, this approach significantly improves production efficiency while reducing manufacturing costs and enhancing adaptability to multi-batch customized production. Practical implementation achieves a 35,190-unit five-day output with notable economic benefits. This research provides a replicable technical framework for intelligent transformation in photovoltaic manufacturing and offers theoretical/practical references for similar discrete manufacturing scenarios.

[16] Guo K, Wan X, Liu L, Gao Z, Yang M. Fault diagnosis of intelligent production line based on digital twin and improved random forest, Applied. Sciences. 2021;11(16):7733. https://doi.org/10.3390/app11167733

[17] Zhang D, Leng J, Xie M, Yan H, Liu Q. Digital twin enabled optimal reconfiguration of the semi-automatic electronic assembly line with frequent changeovers. Robotics and Computer-Integrated Manufacturing. 2022;77:102343. https://doi.org/10.1016/j.rcim.2022.102343

[18] Dai B, Frusque G, Li T, Li Q, Fink O. Smart filter aided domain adversarial neural network for fault diagnosis in noisy industrial scenarios. Engineering Applications of Artificial Intelligence. 2023;126:107202. https://doi.org/10.1016/j.engappai.2023.107202

[19] Ghorvei M, Kavianpour M, Beheshti M, Ramezani A. Spatial graph convolutional neural network via structured subdomain adaptation and domain adversarial learning for bearing fault diagnosis. Neurocomputing. 2023;517:44-61. https://doi.org/10.1016/j.neucom.2022.10.057

[20] Zhang Z, Xue C, Li X, Wang Y, Wang L. A Collaborative Domain Adversarial Network for Unlabeled Bearing Fault Diagnosis. Applied Sciences. 2024; 14(19):9116. https://doi.org/10.3390/app14199116

[21] Ming Z, Tang B, Deng L, Yang Q, Li Q. Digital twin-assisted fault diagnosis framework for rolling bearings under imbalanced data. Applied Soft Computing. 2025;168:112528. https://doi.org/10.1016/j.asoc.2024.112528

[22] Li S, Jiang Q, Xu Y, Feng K, Zhao Z, Sun B, et al. Digital twin-assisted interpretable transfer learning: A novel wavelet-based framework for intelligent fault diagnostics from simulated domain to real industrial domain. Advanced Engineering Informatics. 2024;62:102681. https://doi.org/10.1016/j.aei.2024.102681

[23] Wen H, Guo W, Li X. A novel deep clustering network using multi-representation autoencoder and adversarial learning for large cross-domain fault diagnosis of rolling bearings. Expert Systems With Applications. 2023;225:120066. https://doi.org/10.1016/j.eswa.2023.120066

Question 3: The resolution of the figures in the manuscript could be improved to ensure clarity and enhance the overall presentation of the results.

Reply: We appreciate the reviewer’s valuable feedback regarding the resolution of the figures in our manuscript. To address this concern, we have

---

## [Decision Letter · Decision Letter 1]

8 May 2025

Optimization of Monocrystalline Silicon Photovoltaic Module Assembly Lines Based on Simulation Model

PONE-D-25-10566R1

Dear Dr. Wang,

We’re pleased to inform you that your manuscript has been judged scientifically suitable for publication and will be formally accepted for publication once it meets all outstanding technical requirements.

Kind regards,

Zeashan Hameed Khan, Ph.D.

Academic Editor

PLOS ONE

Additional Editor Comments (optional):

The paper has been significantly improved. Therefore, it can be possibly considered for acceptance.

Reviewers' comments:

Reviewer's Responses to Questions

**Comments to the Author**

Reviewer #1: All comments have been addressed

Reviewer #2: All comments have been addressed

2. Is the manuscript technically sound, and do the data support the conclusions?

Reviewer #1: Yes

Reviewer #2: Partly

3. Has the statistical analysis been performed appropriately and rigorously?

Reviewer #1: Yes

Reviewer #2: Yes

4. Have the authors made all data underlying the findings in their manuscript fully available?

Reviewer #1: No

Reviewer #2: Yes

5. Is the manuscript presented in an intelligible fashion and written in standard English?

Reviewer #1: Yes

Reviewer #2: Yes

Reviewer #1: The concerns from reviewers ahve been well addressed. The quality of this paper has been improved. It cam be accepted now.

Reviewer #2: The second revision addressed key points raised by the reviewers I however suggest a small addition - before finalizing adding a paragraph with the future research recommendations. This can serve as inspiration for other scientists and industry practitioners. While authors do not go deep into the subject it is a successful initial attempt opening doors for further, deeper research which can contribute towards the industry efficiency as well as offer positive impact on the environment (electronic simulation versus physical assets trials).

**Do you want your identity to be public for this peer review?** For information about this choice, including consent withdrawal, please see our Privacy Policy

Reviewer #1: No

Reviewer #2: No

---

## [Editor Report · Acceptance letter]

PONE-D-25-10566R1

PLOS ONE

Dear Dr. Wang,

I'm pleased to inform you that your manuscript has been deemed suitable for publication in PLOS ONE. Congratulations! Your manuscript is now being handed over to our production team.

Kind regards,

on behalf of

Dr. Zeashan Hameed Khan

Academic Editor

PLOS ONE